# iFlood: A Stable and Effective Regularizer

**Yuexiang Xie**[1], **Zhen Wang**[1], **Yaliang Li**[1], **Ce Zhang**[2], **Jingren Zhou**[1], **Bolin Ding**[1]

[1]Alibaba Group, [2]ETH Zürich

{yuexiang.xyx, jones.wz, yaliang.li, jingren.zhou, bolin.ding}@alibaba-inc.com, ce.zhang@inf.ethz.ch

## Abstract

Various regularization methods have been designed to prevent overfitting of machine learning models. Among them, a surprisingly simple yet effective one, called Flooding, is proposed recently, which directly constrains the training loss on average to stay at a given level. However, our further studies uncover that the design of the loss function of Flooding can lead to a discrepancy between its objective and implementation, and cause the instability issue. To resolve these issues, in this paper, we propose a new regularizer, called individual Flood (denoted as iFlood). With instance-level constraints on training loss, iFlood encourages the trained models to better fit the under-fitted instances while suppressing the confidence on over-fitted ones. We theoretically show that the design of iFlood can be intrinsically connected with removing the noise or bias in training data, which makes it suitable for a variety of applications to improve the generalization performances of learned models. We also theoretically link iFlood to some other regularizers by comparing the inductive biases they introduce. Our experimental results on both image classification and language understanding tasks confirm that models learned with iFlood can stably converge to solutions with better generalization ability, and behave consistently at instance-level.

## 1 Introduction

Though overparameterized neural networks have achieved success on a wide range of tasks and applications, it is worth noting that their capacities are sufficient to memorize the entire training data (Zhang et al., 2017a; Arpit et al., 2017), which often leads to an intolerable generalization gap, or in other words, overfitting. To prevent overfitting, many methods have been proposed to regularize how machine learning models fit the training data via introducing additional constraints to control the capacity in effect. For example, $L_1$- or $L_2$- regularizer (i.e., Weight Decay) (Hanson & Pratt, 1989), Early Stopping (Yao et al., 2007), Dropout (Srivastava et al., 2014), Label Smoothing (Szegedy et al., 2016), Confident Penalty (Pereyra et al., 2017), etc (Zhang et al., 2017b; Izmailov et al., 2018; Zheng et al., 2021; Foret et al., 2021; Yang et al., 2020).

Recently, a new method named Flooding (Ishida et al., 2020) is proposed, which controls the extent to which machine learning models fit the training data via directly encouraging the averaged training loss to stay at a given level rather than achieving (near-)zero loss. Formally, Flooding suggests the following loss function $\mathcal{L}_{\text{Flooding}}$ to be minimized:

$$\mathcal{L}_{\text{Flooding}} = \left| \mathcal{L} - b \right| + b, \tag{1}$$

where $\mathcal{L}$ denotes the averaged training loss that is defined over a total of $N$ training instances as $\mathcal{L} = \frac{1}{N} \sum_{i=1}^{N} \mathcal{L}_i$, and $b \geq 0$ is a hyper-parameter called "the flood level" to control the training loss. Designed in such a way, whenever $\mathcal{L}$ is below $b$, the gradients will be negated to increase it, so that $\mathcal{L}$ will stay around $b$ and avoid being (near-)zero. Ishida et al. (2020) suggests to implement Flooding by minimizing $\mathcal{L}_{\text{Flooding}}$ with mini-batch SGD, where $\mathcal{L}$ is estimated over a mini-batch of instances rather than the full data. Although they have pointed out that the objective being optimized by SGD is an upper bound of its desired one, we notice that this gap increases as the batch size decreases, which makes the discrepancy between the objective and implementation of Flooding more serious.

By investigating how machine learning models learned with Flooding behave on the training instances and generalize on the testing ones, we uncover the instability issue of Flooding, where it can lead

to different solutions, and the solutions are inconsistent in their generalization abilities and their behaviors over individuals. Further, we point out that the reason for the instability issue is that Flooding can only guarantee "global convergence"— Flooding encourages the averaged training loss to be sufficiently close to $b$, while having no requirement on the individual losses.

Both the discrepancy and instability issues can be attributed to the design of its loss function, which motivates us to propose a new regularizer in this paper, called individual Flood (denoted as iFlood). The proposed regularizer iFlood defines a loss function $\mathcal{L}_{\text{iFlood}}$ as:

$$\mathcal{L}_{\text{iFlood}} = \frac{1}{N} \sum_{i=1}^{N} \left( \left| \mathcal{L}_i - b \right| + b \right), \tag{2}$$

where $b \geq 0$ is the flood level, $N$ is the size of training sample and $\mathcal{L}_i$ is the loss function defined upon the $i$-th instance. When $b = 0$, $\mathcal{L}$, $\mathcal{L}_{\text{Flooding}}$, and $\mathcal{L}_{\text{iFlood}}$ are equivalent. With $b > 0$, iFlood encourages the model to better fit the under-fitted instances while suppressing the confidence of over-fitted ones.

Although the modification is simple, the proposed new regularizer has the following merits: (1) The design of the loss function of iFlood ensures that it can be optimized by SGD without discrepancy between its objective and implementation. (2) Compared with Flooding, the models learned with iFlood can achieve "local convergence", that is, the individual losses (i.e., $\mathcal{L}_i$s) rather than the averaged loss (i.e., $\mathcal{L}$) are encouraged to be sufficiently close to the specified level $b$, which ensures that the learned models behave consistently over individual instances and produce stable generalization performance. (3) Meanwhile, we theoretically show that the design of iFlood can be intrinsically connected with removing the noise or bias in the training data, making iFlood suitable for a variety of applications to improve the generalization abilities. (4) Moreover, we theoretically compare iFlood with some related works (Szegedy et al., 2016; Pereyra et al., 2017), showing that iFlood discounts the over-confident predictions with less inductive bias.

We conduct extensive experiments on both image classification and language understanding tasks to compare the performance improvements gained by different regularizers, demonstrating the effectiveness of the proposed iFlood. Further, we evaluate the stability[1] of iFlood from several measurements, such as total variation distance and gradient norm. All the experimental results show that, with the "local convergence" suggested by iFlood, the learned models stably converge to solutions with better generalization ability.

## 2 PRELIMINARY

For ease of discussion, we introduce some notations at first. Without loss of generality, we consider a typical classification problem: Given a training dataset $\mathcal{D} = \{(x_i, y_i) | x \in \mathcal{X}, y \in \mathcal{Y}, i = 1, \dots, N\}$ where $\mathcal{X}$ stands for the instance domain, $\mathcal{Y}$ stands for the set of labels, and each instance $(x_i, y_i)$ is independently drawn from an underlying joint distribution $\Pr(X, Y)$, we aim to learn a function $f : \mathcal{X} \to \Delta(\mathcal{Y})$ (i.e., a mapping from the instance domain to the space of probability distributions over the labels) to minimize the generalization error $\mathbb{E}_{(x,y) \sim \Pr(X,Y)} \left[ \mathbf{1}_{y \neq \arg\max_{y'} f(x)_{y'}} \right]$, where $f(x)_{y'}$ denotes the probability of taking the class $y'$.

The function $f$ is often learned by minimizing certain loss function $l(y, f(x))$ (e.g., Cross-Entropy loss). For simplicity, we denote the loss over the $i$-th training instance as $\mathcal{L}_i$ and that over the whole training dataset as $\mathcal{L} = \frac{1}{N} \sum_{i=1}^{N} \mathcal{L}_i$. Note that, besides the classification problem, the regularization methods discussed in this paper are applicable to other machine learning tasks, e.g., regression.

## 3 INDIVIDUAL FLOOD (IFLOOD)

In this section, we first compare iFlood with Flooding from various aspects to demonstrate the advantages of the proposed new regularizer iFlood. Then we provide theoretical analysis about the effect of iFlood for removing the noise or bias in the training data, and connect iFlood with existing regularization methods, such as Label Smoothing (Szegedy et al., 2016) and Confident Penalty (Pereyra et al., 2017).

---

[1]In this paper, "stability" has a different meaning from that used in learning theory (Mohri et al., 2018).

## 3.1 LOCAL CONVERGENCE

Let us revisit how Flooding works first. By applying Flooding, once the original averaged loss $\mathcal{L}$ has approached the flood level, it goes below and above $b$ repeatedly until its convergence. Such process is called "flooding" in Ishida et al. (2020). The loss function of Flooding encourages the averaged loss $\mathcal{L}$ to approach to $b$, thus Flooding pursues the "global convergence" (i.e., $|\mathcal{L} - b| \approx 0$). To learn the model parameters $\theta$ by minimizing $\mathcal{L}_{\text{Flooding}}$ defined in Eq.(1), the gradient of $\mathcal{L}_{\text{Flooding}}$ w.r.t. $\theta$ is in the same direction as that of $\mathcal{L}$ when $\mathcal{L} \geq b$, and in the opposite direction when $\mathcal{L} < b$. To be specific,

$$\nabla_\theta \mathcal{L}_{\text{Flooding}} = \begin{cases} \nabla_\theta \mathcal{L} = \dfrac{1}{N} \sum_{i=1}^{N} \nabla_\theta \mathcal{L}_i, & \text{if } \mathcal{L} \geq b, \\ -\nabla_\theta \mathcal{L} = \dfrac{1}{N} \sum_{i=1}^{N} (-1) \cdot \nabla_\theta \mathcal{L}_i, & \text{if } \mathcal{L} < b. \end{cases} \tag{3}$$

From Eq.(3), we know that whether the original gradient of an instance (i.e., $\nabla_\theta \mathcal{L}_i$) should be negated is determined by the fact that whether the averaged loss $\mathcal{L}$ is below or above the flood level $b$. However, in most practical cases and meanwhile the experiments of Flooding, deep neural networks are trained with mini-batch SGD where $\mathcal{L}$ is estimated from a sampled mini-batch of instances at each step. Thus, whether the gradients should be negated is determined by the averaged training loss estimated over a mini-batch of instances rather than the full data. In this way, the objective being actually optimized by mini-batch SGD is an upper bound of $\mathcal{L}_{\text{Flooding}}$, as pointed out by Ishida et al. (2020). This leads to a discrepancy between the loss function of Flooding and its implementation, which introduces randomness to the choice of whether to negate the gradients and makes the flooding process become random. In this study, we further notice that, due to the property of absolute operator, such discrepancy increases along with the decreasing of batch size.

For iFlood, according to Eq.(2), in a training dataset consisting of $N$ instances, the gradient of $\mathcal{L}_{\text{iFlood}}$ w.r.t. the model parameters $\theta$ can be given as $\nabla_\theta \mathcal{L}_{\text{iFlood}} = \frac{1}{N} \sum_{i=1}^{N} \nabla_\theta |\mathcal{L}_i - b|$. Therefore, for each individual, it is guaranteed to contribute $\frac{1}{N} \nabla_\theta \mathcal{L}_i$ to the aggregated gradient, when $\mathcal{L}_i \geq b$; or to contribute the negation of that, when $\mathcal{L}_i < b$. By applying iFlood, the model parameters are encouraged to walk along the contour of $\mathcal{L} = b$ like what Flooding does, but, in contrast to Flooding, this is achieved by demanding each individual loss $\mathcal{L}_i, \forall i \in \{1, \ldots, N\}$ to be close to $b$. Intuitively speaking, iFlood encourages the model to *better fit the under-fitted instances while suppressing the confidence of over-fitted ones*, according to the given flood level $b$. In a word, iFlood pursues the "local convergence" (i.e., $|\mathcal{L}_i - b| \approx 0, \forall i \in \{1, \ldots, N\}$). We will analyze the advantage of "local convergence" over "global convergence" from the aspect of stability in Section 3.2. For now, the additive property of Eq.(2) ensures the gradients calculated in each SGD an unbiased estimation, which seemingly eliminates the discrepancy between the design and implementation.

Note that when optimized by SGD with batch size of 1, Flooding coincides with iFlood. However, it is unreasonable to set batch size to 1 for training machine learning models in most cases, which is empirically confirmed in Appendix C. For practical batch sizes, the designs of the loss function of iFlood and Flooding are different, as aforementioned, and we will empirically demonstrate those differences in Section 4.

## 3.2 STABILITY

To better understand the differences between Flooding and iFlood, we plot the distributions of individual training losses with Flooding and iFlood in Figure 1, which is produced by training a ResNet18 model on CIFAR-10. Compared with the distribution corresponding to Flooding (see Figure 1a), we can observe that the distribution corresponding to iFlood (i.e., Figure 1b) is much more concentrated, and almost all the instances can be regularized by iFlood to achieve "local convergence" (i.e., $|\mathcal{L}_i - b| \approx 0, \forall i \in \{1, \ldots, N\}$).

Figure 1a confirms that the models learned with Flooding can achieve "global convergence" (i.e., $|\mathcal{L} - b| \approx 0$). The inconsistent in behaviors over individuals achieved by Flooding can causes the instability of Flooding, namely that Flooding leads to various solutions with different generalization abilities. Let's consider the following cases: One learned model achieves $\mathcal{L}_i = b$ for every training instance, and the other learned model achieves $\mathcal{L}_i = 0$ on half of the whole training instances while

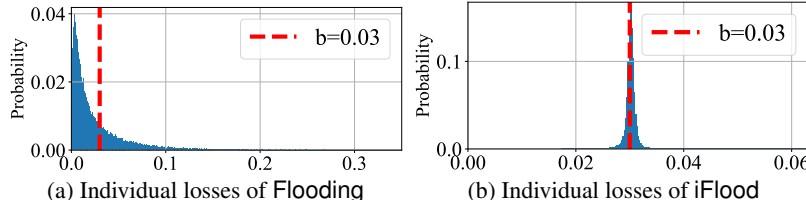

Figure 1: The distributions of individual losses of Flooding (a) and iFlood (b). The read dotted lines represent the flood level $b = 0.03$. Best viewed with color.

$\mathcal{L}_i = 2b$ on the rest ones. Obviously, these two learned models have achieved the same "global convergence" that $\mathcal{L} - b = 0$, but they behave inconsistently over the (training) individuals. Moreover, when $b$ is taken to be a level where $\mathcal{L}_i = b$ corresponds to being correctly classified and $\mathcal{L}_i = 2b$ corresponds to being mis-classified, the generalization ability of the former surpasses that of the latter in most cases. The analysis of these cases sheds light on the reason for Flooding's instability. Albeit the learning dynamics would not lead to these extreme cases in practice, we experimentally observe that, over a considerable portion of instances, the individual losses produced by a model learned with Flooding are far away from $b$, as Figure 1a shows.

To address this issue, iFlood is introduced to achieve "local convergence", which ensures the learned models behave more consistently than those models learned with Flooding. In this way, iFlood can achieve a stable boost of generalization ability. We will provide quantitative analysis to demonstrate the advantages of iFlood on stability in Section 4.

### 3.3 THEORETICAL ANALYSIS

It has been noticed by Ishida et al. (2020) that a larger $b$ is in favor, when the training data contain noisy labels. Intuitively, $b$ controls how confident the model should trust the training data. Inspired by this observation, it is worth discussing (1) what is the meaning of $b$ in iFlood; (2) when and why do the models learned with iFlood benefit from the regularization. In the rest of this section, we look into these questions under two settings—noisy labels and biased sample, which are commonly considered in both academia and industry.

#### 3.3.1 NOISY LABELS

For many real-world applications, the observed labels are polluted due to certain reasons (Angluin & Laird, 1988; Patrini et al., 2016; Thulasidasan et al., 2019), e.g., the mistakes made by annotators, the perturbation of some attackers, etc. Here, we consider a widely adopted setting where a noisy training sample $\mathcal{D}_{\text{noisy}} = \{(x_i, z_i) | x_i \in \mathcal{X}, z_i \in \mathcal{Y}, i = 1, \ldots, N\}$ is given, with each $(x_i, z_i)$ being independently drawn from a distribution $\Pr(X, Z)$ in the following process: first, an instance $(x, y)$ is drawn from the underlying joint distribution $\Pr(X, Y)$; then, $y$ is perturbed to be $z$, taking an incorrect label $y'$ with probability $\alpha, 0 \leq \alpha \leq 1$, or keeping the correct label $y$ with probability $(1 - \alpha)$, where the incorrect label is drawn from a uniform distribution over $\mathcal{Y}\backslash\{y\}$. This generation process implies $Z \perp\!\!\!\perp X | Y$ and a conditional probability distribution $\Pr(Z|Y)$ with $\forall y \in \mathcal{Y}, \Pr(Z = y|y) = 1 - \alpha$ and $\forall y' \neq y, \Pr(Z = y'|y) = \frac{\alpha}{|\mathcal{Y}|-1}$.

Given $\mathcal{D}_{\text{noisy}}$, suppose the data generation process is known, we can learn a model parameterized with $\theta$ by minimizing its negative log-likelihood:

$$-\log \Pr(z_{1:N}|x_{1:N}, \theta) = -\log \prod_{i=1}^{N} \Pr(z_i|x_i, \theta) = -\sum_{i=1}^{N} \log \sum_{y \in \mathcal{Y}} \Pr(y_i, z_i|x_i, \theta)$$

$$= -\sum_{i=1}^{N} \log \sum_{y \in \mathcal{Y}} \Pr(z_i|y_i) \Pr(y_i|x_i, \theta) = -\sum_{i=1}^{N} \log\{(1-\alpha)\Pr(Y = z_i|x_i, \theta) + \frac{\alpha}{|\mathcal{Y}|-1} \sum_{y \neq z_i} \Pr(y|x_i, \theta)\}$$

$$= -\sum_{i=1}^{N} \log\{(1-\alpha)\Pr(Y = z_i|x_i, \theta) + \frac{\alpha}{|\mathcal{Y}|-1}(1 - \Pr(Y = z_i|x_i, \theta))\},$$

$$\tag{4}$$

where we denote the cardinality of $\mathcal{Y}$ as $|\mathcal{Y}|$. However, when we have no idea about the underlying data generation process, the RHS of Eq.(4) is unknown to us.

If $\mathcal{D}_{\text{noisy}}$ is directly adopted as training data without denoising, $\theta$ will be learned by minimizing the Cross-Entropy loss: $\mathcal{L} = \frac{-1}{N} \sum_{i=1}^{N} \sum_{y \in \mathcal{Y}} \mathbf{1}_{z_i = y} \log \Pr(Y = y | x_i, \theta)$. In this case, the learned model tends to generalize poorly, as it has been misled by the noisy sample and learns a distribution different from the actual joint distribution $\Pr(X, Y)$.

To explore the usage of iFlood in denoising, we first analyze Eq.(4) and present the following proposition about it.

**Proposition 1.** *The negative log-likelihood (i.e., Eq.(4)) is upper bounded by:*

$$N \log(|\mathcal{Y}| - 1) - \sum_{i=1}^{N} \{(1 - \alpha) \log \Pr(Y = z_i | x_i, \theta) + \alpha \log(1 - \Pr(Y = z_i | x_i, \theta))\}, \qquad (5)$$

*which is minimized when* $\Pr(Y = z_i | x_i, \theta) = 1 - \alpha, i = 1, \ldots, N$.

*Proof.* Since $-\log(\cdot)$ is monotonically decreasing and $|\mathcal{Y}| \geq 2$, Eq.(4) is upper bounded by: $-\sum_{i=1}^{N} \log\{\frac{1-\alpha}{|\mathcal{Y}|-1} \Pr(Y = z_i | x_i, \theta) + \frac{\alpha}{|\mathcal{Y}|-1}(1 - \Pr(Y = z_i | x_i, \theta))\}$ within its domain. This function is further upper bounded by Eq.(5) with Jensen's inequality. By checking the derivatives of Eq.(5), we know that it is minimized when $\Pr(Y = z_i | x_i, \theta) = 1 - \alpha, i = 1, \ldots, N$. $\qquad \square$

When we regularize the Cross-Entropy loss defined over $\mathcal{D}_{\text{noisy}}$ with iFlood, it becomes:

$$\mathcal{L}_{\text{iFlood}} = \frac{1}{N} \sum_{i=1}^{N} \{| - \sum_{y \in \mathcal{Y}} \mathbf{1}_{z_i = y} \log \Pr(Y = y | x_i, \theta) - b| + b\}. \qquad (6)$$

Once we specify $b = -\log(1 - \alpha)$, $\mathcal{L}_{\text{iFlood}}$ encourages every $\Pr(Y = z_i | x_i, \theta), i = 1, \ldots, N$ to take $\exp(-b) = 1 - \alpha$, which minimizes an upper bound (Eq.(5)) of the actual negative log-likelihood (Eq.(4)). In this way, without knowing how the observed sample becomes noisy beforehand, $\mathcal{L}_{\text{iFlood}}$ can still serve as a surrogate function for the unknown negative log-likelihood (Eq.(4)) of interest, extending the usage of iFlood to denoising.

### 3.3.2 BIASED SAMPLE

The second setting to explore iFlood is when the collected training data does not follow the probability distribution of interest. Such a data sample is often named biased sample, which is ubiquitously observed in real-world applications (Abdollahpouri et al., 2019; Kowald et al., 2020). For instance, in recommendation systems, there exists popularity bias (Abdollahpouri et al., 2019; Kowald et al., 2020), and in face recognition and object recognition, data samples are biased regarding the backgrounds or the illumination conditions (Kortylewski et al., 2019; Barbu et al., 2019). Again, we represent an observed label by $Z$ and denote such a biased sample as $\mathcal{D}_{\text{biased}} = \{(x_i, z_i) | x_i \in \mathcal{X}, z_i \in \mathcal{Y}, i = 1, \ldots, N\}$. The biased sample setting can be formally given as: The observed distribution of input, denoted as $\Pr'(X)$, is different from the ground-truth one $\Pr(X)$, which leads to the joint distribution $\Pr(X, Z) \neq \Pr(X, Y)$ and thus model tends to learn a biased conditional distribution $\Pr(Z|X)$ rather than the underlying one $\Pr(Y|X)$.

Models learned from a biased sample cannot generalize well due to the distribution drift. Luckily, we can apply iFlood to debiase. In most cases, although we have no idea about the analytic form of $\Pr(Z|X, Y)$, it is reasonable to assume and validate some properties about $Z|X$ that $\Pr(Z|X, Y)$ implies. With such properties, iFlood is able to recover the unbiased label. Formally, let's assume that according to distributions $\Pr(Z|X, Y)$ and $\Pr(X, Y)$ which generate $\mathcal{D}_{\text{biased}}$, the difference between observed labels $Z|X$ and underlying labels $Y|X$ follows some specific distribution:

$$l(Z|X, Y|X) \sim \text{Lap}(\cdot | \mu, \lambda), \qquad (7)$$

where $l$ denotes the adopted loss function, and $\text{Lap}(\cdot | \mu, \lambda)$ denotes the Laplace distribution with parameters $\mu \geq 0, \lambda > 0$. With such knowledge, the negative log-likelihood of observing the biased sample $\mathcal{D}_{\text{noisy}}$ is as follows:

$$-\frac{1}{N} \sum_{i=1}^{N} \log \Pr[l(z_i, f(x_i))] = -\frac{1}{N} \sum_{i=1}^{N} \log\{\frac{1}{2\lambda} e^{-\left|\frac{l(z_i, f(x_i)) - \mu}{\lambda}\right|}\} \propto \frac{1}{N} \sum_{i=1}^{N} |l(z_i, f(x_i)) - \mu|. \qquad (8)$$

This happens to be in the same form as $\mathcal{L}_{\text{iFlood}}$. Thus, learning a model from the biased sample $\mathcal{D}_{\text{biased}}$ via iFlood with $b = \mu$ leads to the same solution as learning from the corresponding unbiased sample.

### 3.4 Comparing iFlood with Other Related Works

There exist some regularization methods that prevent a model from overfitting by suppressing its over-confident predictions, such as Label Smoothing (denoted as LS) (Szegedy et al., 2016) and Confident Penalty (denoted as CP) (Pereyra et al., 2017). Following Meister et al. (2020), these methods can be summarized as follows:

$$\mathcal{L}_{\text{LS}} = \mathcal{L} + \beta D_{KL}(u\|p), \qquad \mathcal{L}_{\text{CP}} = \mathcal{L} + \beta D_{KL}(p\|u), \tag{9}$$

where $\beta$ is the regularization strength, $D_{KL}(\cdot\|\cdot)$ represents the KL divergence, $p$ is the predicted probability distribution over labels, and $u$ is a prior of the label distribution. From Eq.(9) we can see that Label Smoothing and Confident Penalty add a regularization term to the loss function $\mathcal{L}$ to encourage the distribution of predicted probability to be close to the prior. Without additional domain knowledge, the uniform distribution is often adopted as the prior, which is expected to prevent the peaked distributions (i.e., the over-confident predictions) and lead to a better generalization.

Compared with Label Smoothing and Confident Penalty, iFlood performs similarly but has less assumption on the prior of the label distribution. iFlood prevents the model from becoming over-confident on the observed training sample via discounting its prediction probability over the observed label, where the extent of discount is controlled by the flood level $b$. On the other hand, iFlood does not make any assumption on how to distribute the reserved confidence.

Formally, we compare iFlood with Label Smoothing under the setting where Cross-Entropy loss is considered and a uniform prior is adopted by Label Smoothing. In this case, the predicted probability distribution $p$ (over labels) that minimizes the objective of Label Smoothing (Eq.(9)) is exactly one of the distributions that minimize the loss function of iFlood. As the regularization is posed at instance-level in both methods, we show this relationship by checking the minimizer of the objective of Label Smoothing over any $(x, y) \in \mathcal{D}$:

$$l(y, f(x)) + \beta D_{\text{KL}}(u\|f(x)) = l(y, p) + \beta D_{\text{KL}}(u\|p) = -\log(p_y) - \frac{\beta}{|\mathcal{Y}|}\log(p_y) - \frac{\beta}{|\mathcal{Y}|}\sum_{y' \neq y}\log(p_{y'}) \tag{10}$$

$$\geq -\Big[\frac{|\mathcal{Y}| + \beta}{|\mathcal{Y}|}\log(p_y) + \frac{(|\mathcal{Y}| - 1)\beta}{|\mathcal{Y}|}\log\big(\frac{1 - p_y}{|\mathcal{Y}| - 1}\big)\Big] = -\Big[\frac{|\mathcal{Y}| + \beta}{|\mathcal{Y}|}\log(p_y) + \frac{(|\mathcal{Y}| - 1)\beta}{|\mathcal{Y}|}\log(1 - p_y) + c\Big], \tag{11}$$

where $p$ denotes the probability distribution over labels predicted by $f$, $c$ is a constant, and $|\mathcal{Y}|$ denotes the cardinality of $\mathcal{Y}$. On one hand, Eq.(11) is a tight lower bound of Eq.(10), where equality is established when $p$ satisfies that $\forall y' \neq y, p_{y'} = \frac{1 - p_y}{|\mathcal{Y}| - 1}$. On another hand, derivative of Eq.(11) shows that it (as well as Eq.(10)) can be minimized at $p_y = \frac{|\mathcal{Y}| + \beta}{|\mathcal{Y}|(1 + \beta)}$. Thus, the objective of Label Smoothing encourages the learned function $f$ to predict a distribution $p$ that satisfies both of the above conditions. Meanwhile, when we specify iFlood with $b = \log(\frac{|\mathcal{Y}|(1 + \beta)}{|\mathcal{Y}| + \beta})$, according to Eq.(2), it encourages the learned function $f$ to predict a distribution $p$ that satisfies $p_y = \exp(-b) = \frac{|\mathcal{Y}| + \beta}{|\mathcal{Y}|(1 + \beta)}$. Based on the above analysis, we see that iFlood discounts the prediction confidence over the observed label to the same level as that of Label Smoothing, without encouraging the reserved confidence to be equally distributed to other labels. When it is hard to obtain some prior knowledge about the label distribution, iFlood can provide a less-biased and more flexible regularization, compared with Label Smoothing. Similar analysis and conclusion can be applied to Confident Penalty.

## 4 Experiments

In this section, we conduct a series of experiments to demonstrate the effectiveness of iFlood, with the aim to answer the following questions: **Q1**: Does iFlood provide larger boost of generalization ability, compared with existing regularization methods on benchmark datasets? **Q2**: Compared to Flooding, can iFlood stably converge to solutions with better generalization ability? **Q3**: Can iFlood address the noisy label issue well in practice?

Table 1: Accuracy (%) comparison on benchmark datasets.

| Regularizer | CIFAR-10 | CIFAR-100 | SVHN | ImageNet | SST-2 | QQP | QNLI |
|---|---|---|---|---|---|---|---|
| Unregularized | 94.59 | 78.24 | 96.94 | 77.24 | 91.88 | 90.40 | 90.79 |
| Label Smoothing | 94.78 | 77.32 | 97.06 | 77.44 | 91.63 | 91.06 | 91.35 |
| Confident Penalty | 94.61 | 78.28 | 97.01 | 77.28 | 91.88 | 91.10 | 91.53 |
| Flooding | 94.58 | 78.63 | 96.98 | 77.25 | 91.86 | 91.14 | 91.43 |
| iFlood (ours) | **94.95** | **79.06** | **97.16** | **77.58** | **92.09** | **91.22** | **91.64** |

## 4.1 SETTINGS

**Datasets.** We consider both image classification and language understanding tasks. For image classification, we use CIFAR-10, CIFAR-100 (Krizhevsky et al., 2009), SVHN (Netzer et al., 2011), and ImageNet (Russakovsky et al., 2015). For language understanding, we adopt the General Language Understanding Evaluation (GLUE) benchmark (Wang et al., 2019), and report the experimental results on SST-2, QQP, and QNLI. The details of these datasets and more experimental results on GLUE benchmark can be found in Appendix A and D.2 respectively.

**Baselines.** We compare iFlood with the following baselines: (1) Flooding (Ishida et al., 2020); (2) Label Smoothing (Szegedy et al., 2016); (3) Confident Penalty (Pereyra et al., 2017); (4) Unregularized, denotes that the models are learned without Flooding, iFlood, Label Smoothing and Confident Penalty. Data augmentation (e.g., random crop (Krizhevsky et al., 2012) and horizontal flip (Simonyan & Zisserman, 2015)) have been necessary plug-ins of the learning procedure on image classification datasets, we do not remove them in our experiments. As the basic regularizers, $L_2$-regularization is used in both image classification and language understanding tasks, and Dropout is adopted in language understanding task.

**Implementation details.** On the image classification datasets, we consider training ResNet18 (He et al., 2016) for CIFAR-10, CIFAR-100 and SVHN, and training ResNeXt50 (Xie et al., 2017) for ImageNet, adopting momentum SGD as the learning procedure to be regularized. We train ResNet18 for 300 epochs with 128 as the batch size. The learning rate is initialized as 0.1 and decays (multiplied by 0.2) at the 80-th, 160-th and 200-th epochs. As for ResNeXt50, we train it for 90 epochs with 256 as the batch size. The learning rate is initialized as 0.1 and decays (multiplied by 0.1) at the 30-th, and 60-th. On the language understanding datasets, we consider fine-tuning BERT (Devlin et al., 2019) via Adam (Kingma & Ba, 2015) as the learning procedure to be regularized. We adopt the pre-trained BERT model provided by huggingface (Wolf et al., 2020) and fine-tune it on the target datasets. The number of epochs is tuned among $\{3, 4, 5\}$, the batch size is 16, the learning rate is tuned among $\{2e-5, 5e-5\}$, and the dropout rate is 0.1. For Flooding and iFlood, the flood level $b$ is tuned in the range of $[0.10, 0.50]$ via grid seach with 0.05 as the step size for ImageNet, and tuned in the range of $[0.01, 0.10]$ via grid search with 0.01 as the step size for other datasets.

Hyper-parameter spaces considered for baseline methods can be found in the Appendix B. All models are implemented using PyTorch (Paszke et al., 2019) and trained on NVIDIA GeForce GTX 1080 Ti or Tesla V100 GPUs. For fair comparison, we search for the optimal configuration of hyper-parameters for each method. Then we run each method for 5 times using its optimal configuration and report the averaged results, reducing the randomness in the comparison.

## 4.2 IMPROVEMENTS ON GENERALIZATION ABILITY (Q1)

We compare iFlood with the baseline methods on benchmark datasets to demonstrate its effectiveness. The experimental results are summarized in Table 1. Obviously, with appropriate configurations (i.e., the strength of regularization), all these regularization methods can improve the performance of the learned model compared to unregularized case, on most of the datasets. This phenomenon can be explained by the fact that, in this experiment, the number of parameters to be estimated is larger than the size of training sample, which makes regularization indispensable. We can also observe that, on both the image classification and language understanding datasets, iFlood provides larger improvements of the performance than those of the baseline methods. This confirms the effectiveness of iFlood in improving generalization ability of learned models, which is brought by the design of iFlood, encouraging the learned model to better fit under-fitted instances while suppressing the confidence on overfitted ones. We also conduct experiments on large-scale dataset *Criteo* and report the results in Appendix D.1, which confirms the effectiveness of iFlood.

Table 2: Comparison of performance variances. The relative changes in terms of "Unregularized" cases are also reported.

| Dataset | Regularizer | The std. of test acc. | max $D_{\text{TV}}$ | avg $D_{\text{TV}}$ |
|---------|-------------|----------------------|---------------------|---------------------|
| CIFAR-10 | Unregularized | 0.16% | 0.0004 | 0.0003 |
| | Flooding | 0.29% (+0.13%) | 0.0237 (×59.25) | 0.0221 (×73.67) |
| | iFlood | 0.19% (+0.03%) | 0.0068 (×17.00) | 0.0053 (×17.67) |
| CIFAR-100 | Unregularized | 0.29% | 0.0029 | 0.0028 |
| | Flooding | 0.36% (+0.07%) | 0.0241 (×8.31) | 0.0236 (×8.43) |
| | iFlood | 0.22% (-0.07%) | 0.0167 (×5.76) | 0.0165 (×5.89) |
| SST-2 | Unregularized | 0.29% | 0.0089 | 0.0084 |
| | Flooding | 0.37% (+0.08%) | 0.0205 (×2.30) | 0.0193 (×2.30) |
| | iFlood | 0.21% (-0.08%) | 0.0103 (×1.16) | 0.0099 (×1.18) |
| QQP | Unregularized | 0.12% | 0.0220 | 0.0173 |
| | Flooding | 0.14% (+0.02%) | 0.0376 (×1.71) | 0.0321 (×1.86) |
| | iFlood | 0.12% (+0.00%) | 0.0223 (×1.01) | 0.0175 (×1.01) |

It is worth pointing out that, iFlood can work well with some other regularization methods, such as weight decay. Under the scenario of over-parameterized neural networks, near-zero training loss is achievable regardless of the regularization constraints posed upon the model parameters. Therefore, when cooperated with weight decay, iFlood controls the ultimate extent to which such models fit the training data, while weight decay affects learning dynamics (Golatkar et al., 2019).

## 4.3 STABILITY OF IFLOOD (Q2)

In Section 3, we provide analysis about the instability of Flooding, which further motivates us to propose iFlood. In this section, we conduct experiments to show that, compared to Flooding, iFlood can stably converge to solutions with better generalization ability.

**Variances.** We use the standard deviation (denoted as "std.") of test accuracy to measure the differences of the generalization abilities among the learned models. Beseides, we adopt the total variation distance (denoted as $D_{\text{TV}}$) to measure the difference between any two learned models, which can be estimated on the training sample as:

$$D_{\text{TV}}^{\mathcal{D}}(f,g) = \frac{1}{2N} \sum_{i=1}^{N} \sum_{y \in \mathcal{Y}} \left| f(\boldsymbol{x}_i)_y - g(\boldsymbol{x}_i)_y \right|, \tag{12}$$

where $f$ and $g$ denote two learned models. We estimate the distance for every pair of the ten models and report both the maximum and the averaged values of them. The experimental results are summarized in Table 2.

The results show that the test accuracy of models learned with iFlood vary less than that of models learned with Flooding, with the largest difference between them could be 0.16%. The less variance in generalization ability demonstrates the stability of iFlood. Meanwhile, from Table 2 we can observe that, the variance of iFlood is much smaller than that of Flooding (e.g., nearly 24% - 70% of Flooding from the aspect of avg $D_{\text{TV}}$). Further, the comparison of total variation distance $D_{\text{TV}}$ between Flooding and iFlood is consistent with the individual losses shown in Figure 1, which supports our idea: Under the individual-level constraint, iFlood can achieve "local convergence" and converge to solutions with better generalization ability, in a stabler manner than Flooding.

**The norm of gradients.** To further compare the stability between Flooding and iFlood, we train a ResNet18 on CIFAR-10 (Figure 2a) and CIFAR-100 (Figure 2b), and monitor the $L_1$ norm of the gradients at each epoch. From the figure, we can observe that, during the training course of Flooding, the norm of gradients is larger than that of iFlood by a noticeable margin, and the gap has not been filled until the end of training. It implies that even at the end of training, the model parameters learned with Flooding are changed more significantly than those of iFlood, which could cause the instability of learned models.

**The effect of $b$.** We further study the effect of flood level $b$ via training a ResNet18 on CIFAR-10, and report the training accuracies of both Flooding and iFlood with varied $b$ in Figure 2c. From the

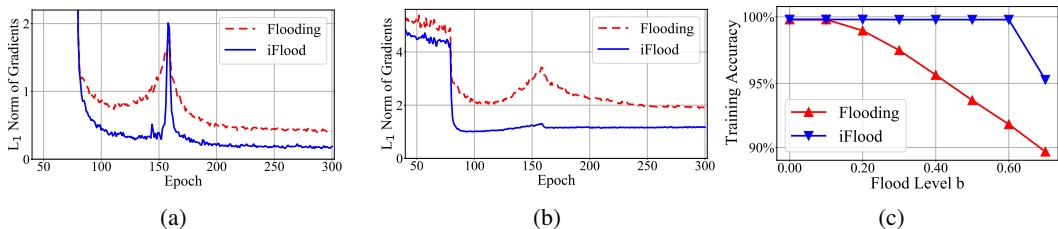

Figure 2: The comparisons between Flooding and iFlood. (a) The $L_1$ norm of gradients on CIFAR-10; (b) The $L_1$ norm of gradients on CIFAR-100; (c) The training accuracy on CIFAR-10 w.r.t. various flood level $b$. Best viewed with color.

figure we can observe that, as the flood level $b$ increases, the training accuracy of Flooding drops drastically, while that of iFlood stays at the same level until $b > 0.60$. This result implies that iFlood is much more robust w.r.t. the hyper-parameter $b$ compared to Flooding. Since Flooding just encourages the averaged loss to be close enough to $b$ but has no requirement on the individual losses, some instances stay under-fitted regrading the flood level $b$. When $b$ is taken to be a relatively large value, the training instances whose individual losses are larger than $b$ might become under-fitted or even mis-classified. Such instances might degrade the model's generalization ability (Belkin et al., 2019). As iFlood encourages the model to fit every individual instance to the same extent (i.e., $|\mathcal{L}_i - b| \approx 0$), it can overcome such under-fitted instance issues via achieving "local convergence".

## 4.4 Effectiveness in Denoising (Q3)

In this section, we instantiate the noise label setting to evaluate the effectiveness of iFlood in denoising. Following the setting discussed in Section. 3.3.1, the polluted version of CIFAR-10 and SST-2 training datasets are constructed. We train ResNet18 model on the polluted CIFAR-10 dataset and train BERT model on the polluted SST-2 dataset, with Flooding or iFlood applied as the regularizer for denoising. More implementation details can be found in the Appendix E.

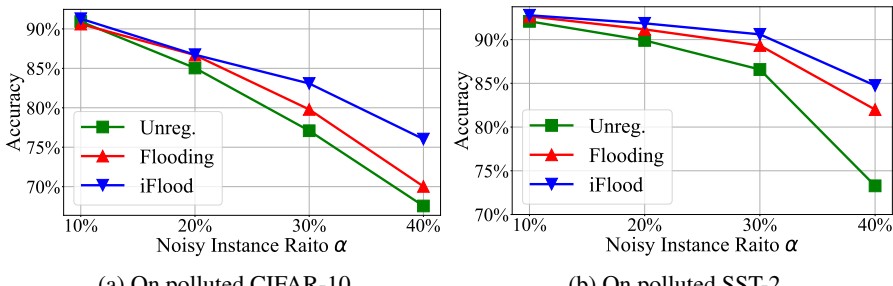

(a) On polluted CIFAR-10                    (b) On polluted SST-2

Figure 3: Performance comparison on datasets polluted with noisy labels.

Results are plotted in Figure 3. It can be observed that, on both polluted CIFAR-10 and polluted SST-2 datasets, models learned with iFlood generalize much better than those with Flooding and those without any regularization. Further, the advantages of iFlood become more significant as the ratio of noisy instance $\alpha$ increases. These experimental results confirm our theoretical analysis in Section 3.3.1 that iFlood, as a regularizer, has the capability to denoise data, so that the regularized models can achieve better generalization performances, even when they are trained on noisy datasets.

## 5 Conclusions

In this paper, we uncover the instability issue of a recently proposed regularization method Flooding. Our analysis on its causes motivates us to propose a new regularizer iFlood, which encourages the individual losses to approach a specified level. Experimental results on a variety of tasks show that the models learned with iFlood produce stabler generalization improvement. We also theoretically show that, the objective of iFlood over a noisy or biased training sample can serve as a surrogate or even match the maximum likelihood estimator applied to a clean sample, which is also confirmed by our experimental results. Further, we theoretically compare iFlood with some other regularization methods and find that iFlood discounts the over-confident predictions with less inductive bias injected.

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

# A    DATASETS

We conside both image classification and language understanding tasks, and adopt 7 benchmark datasets for evaluation, including CIFAR-10[2], CIFAR-100[3], SVHN[4], ImageNet[5], SST-2[6], QQP[7] and QNLI[8]. The statistics of the datasets are summarized in Table 3.

Table 3: The statistics of benchmark datasets.

|  | Dataset | # Train | # Test |
|---|---|---|---|
| Image Classification | CIFAR-10 | 50,000 | 10,000 |
|  | CIFAR-100 | 50,000 | 10,000 |
|  | SVHN | 73,257 | 26,032 |
|  | ImageNet | 1,281,167 | 50,000 |
| Language Understanding | SST-2 | 67,349 | 872 |
|  | QQP | 363,846 | 40,430 |
|  | QNLI | 104,743 | 5,463 |

# B    DETAILS OF HYPERPARAMETER OPTIMIZATION

We randomly split the training data into training and validation sets with the proportion of 9:1, and apply grid search on the validation dataset for hyperparameter optimization (HPO). We adopt the optimal configuration provided by HPO to train and evaluate each method 5 times, alleviating the impact of randomness.

Specifically, for Label Smoothing (Szegedy et al., 2016), we tune the smoothing parameter value $\epsilon$ among $\{0.05, 0.1, 0.2\}$, and adopt a uniform distribution as the prior of label distribution for interpolation. For Confident Penalty (Pereyra et al., 2017), following the original paper, the strength of the confidence penalty $\beta$ is tuned among $\{0.1, 0.5, 1.0, 2.0\}$.

# C    SGD WITH VARIOUS BATCH SIZE

When optimized by SGD with batch size of 1, Flooding coincides with iFlood. However, it is unreasonable to set batch size to 1 for training practical machine learning models, according to the literature and our experimental results shown in Figure 4.

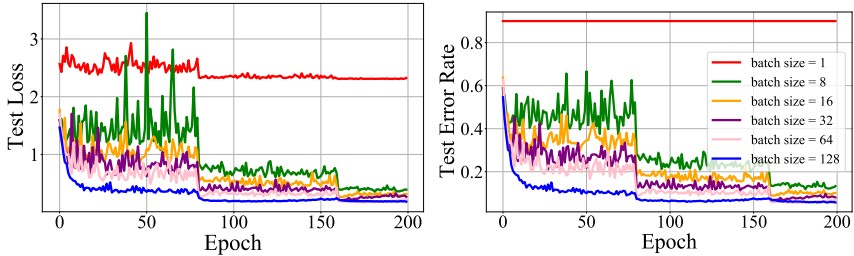

Figure 4: ResNet18 with Flooding on CIFAR-10.

As mentioned in (Ishida et al., 2020), the objective being optimized by SGD is an upper bound of its desired objective, where we notice that the gap between these two objectives increases w.r.t. the

---

[2] https://www.cs.toronto.edu/ kriz/cifar.html

[3] https://www.cs.toronto.edu/ kriz/cifar.html

[4] http://ufldl.stanford.edu/housenumbers/

[5] https://image-net.org/challenges/LSVRC/2012/

[6] https://nlp.stanford.edu/sentiment/index.html

[7] https://data.quora.com/First-Quora-Dataset-Release-Question-Pairs

[8] https://rajpurkar.github.io/SQuAD-explorer/

decreasing of batch size. As for the practical batch sizes, the design of the loss function of iFlood and Flooding are different, and we have shown iFlood outperforms Flooding in Section 4.2.

# D    Experiments on Large-scale Dataset and GLUE Benchmark

## D.1    Large-scale Dataset Criteo

To demonstrate the effectiveness of the proposed iFlood, we train an MLP for click-through rate prediction on *Criteo*, a real-world advertisement dataset including around 45 million instances. The AUC of "Unregurized" v.s. Flooding v.s. iFlood are 78.08% v.s. 78.41% v.s. 79.14%. According to literature, a **0.001-level** improvement in offline AUC evaluation makes a significant difference. Thus these results confirm the effectiveness of iFlood on large-scale real-world datasets.

## D.2    GLUE benchmark

Table 4: Accuracy (%) comparison on GLUE benchmark.

| Dataset | Metric | Unregularized | Label Smoothing | Confident Penalty | Flooding | iFlood |
|---|---|---|---|---|---|---|
| CoLA | Matthews Corr. | 56.82 | 56.72 | 56.85 | 56.53 | **57.87** |
| SST-2 | Accuracy | 91.88 | 91.63 | 91.88 | 91.86 | **92.09** |
| MRPC | Accuracy | 84.46 | 84.36 | 84.90 | 83.64 | **85.25** |
| STS-B | Pearson Corr. | 89.00 | - | - | 89.37 | **89.46** |
| QQP | Accuracy | 90.40 | 91.06 | 91.10 | 91.14 | **91.22** |
| MNLI | Accuracy | **83.22** | 83.43 | 83.12 | **83.22** | **83.22** |
| QNLI | Accuracy | 90.79 | 91.35 | 91.53 | 91.43 | **91.64** |
| RTE | Accuracy | 65.49 | 66.28 | 66.50 | 65.05 | **67.29** |
| WNLI | Accuracy | **56.34** | **56.34** | **56.34** | **56.34** | **56.34** |

The experimental results on GLUE benchmark are shown in Table 4[9], from which we can observe that iFlood outperforms other baseline methods by a noticeable margin. These experimental results are consistency with those reported in Table 1, which confirm the effectiveness of iFlood in improving generalization ability of learned models.

# E    Datasets Construction in Noisy Label

Following the setting discussed in noisy label (Section 3.3.1), the polluted version of CIFAR-10 and SST-2 datasets are constructed as follow: We randomly choose a proportion of the original training instances according to $\alpha(0 < \alpha < 1)$, and then pollute the label of each chosen instance by uniformly picking one from its corresponding incorrect classes. We enumerated the noise instance ratio $\alpha$ with 0.1, 0.2, 0.3 and 0.4. Note that the test instances are kept clean to reflect the real-world scenarios. We search for the optimal flood level $b$ in the range of $[0.10, 0.50]$ via grid search with 0.05 as the step size for both Flooding and iFlood. The experimental results can be found in Section 4.4.

# F    Additional Experiments

## F.1    Low-frequency Component v.s. High-frequency Component

Inspired by previous study (Wang et al., 2020), we conduct experiments to show how model learned with different regularizers reacts to different levels of details of the data (e.g., the low-frequency component and high-frequency component of images). To be specific, for each instance in the training data, we decompose the data into low-frequency component and high-frequency component w.r.t. different radius thresholds $r$ via applying Fourier transform and inverse Fourier transform. Then we train a ResNet-18 on CIFAR-10 using the raw training data, and evaluate the model on both

---

[9]Label Smoothing and Confident Penalty are not suitable to be adopted on STS-B, since the output dimension is 1.

low-frequency component and high-frequency component. More details can be referred to Section 3.1 in Wang et al. (2020).

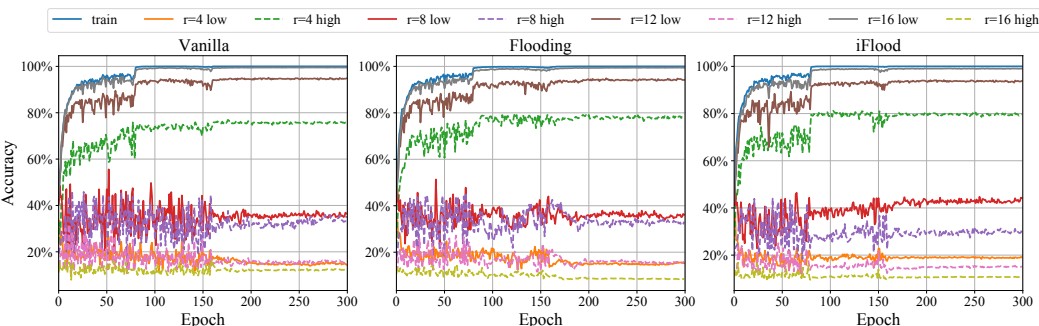

Figure 5: How models learn with low-frequency component and high-frequency component.

The experimental results are illustrated in Figure 5, where $r = 4/8/12/16\ low$ denotes the low-frequency component and $r = 4/8/12/16\ high$ denotes the high-frequency component. From these experimental results we can conclude that model performs better on low-frequency component than high-frequency component when $r = 8/12/16$, but worse when $r = 4$, which are consistency with the results in Wang et al. (2020) (note that BatchNorm is adopted). Compared to Flooding and Vanilla, we can observe that model learned with iFlood catches more low-frequency component (e.g., $r = 4/8\ low$) and less high-frequency component (e.g., $r = 8/16\ high$), which confirms the effectiveness of iFlood in improving the generalization ability of model since low-frequency component is much more generalizable than high-frequency component (Wang et al., 2020).

## F.2 THE NORM OF GRADIENTS

We train a ResNet18 on CIFAR-10 and CIFAR-100, and monitor the $L_1$ norm of the gradients at each epoch. The experimental results shown in Figure 6a and 6b confirm that: (1) The design of the loss function of Flooding leads to the instability issue, which is supported by the phenomenon that the norm of gradients is larger than other methods by a noticeable margin; (2) The design of the loss function of iFlood brings the merit of significantly reducing the gap of the norm of gradients between Flooding and other methods.

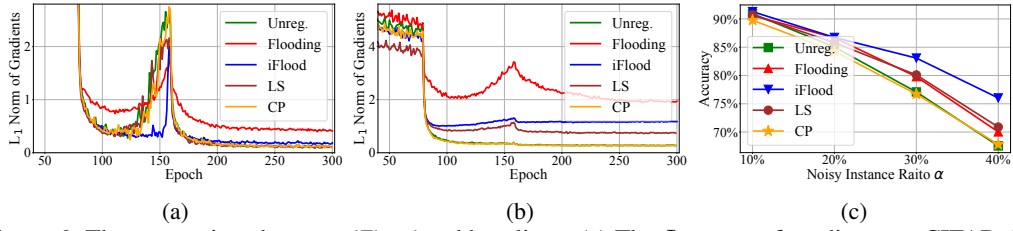

Figure 6: The comparison between iFlood and baselines. (a) The $L_1$ norm of gradients on CIFAR-10; (b) The $L_1$ norm of gradients on CIFAR-100; (c) Performance comparison on polluted CIFAR-10.

## F.3 EFFECTIVENESS IN DENOISING

We train a ResNet-18 on polluted CIFAR-10 to evaluate the effectiveness of iFlood in denoising. The generation process of polluted CIFAR-10 can be referred to Appendix E, and the experimental results are shown in Figure 6c. From the figure we can observe that, models learned with iFlood outperform those with other regularizers by a noticeable margin, and the advantages of iFlood become more significant as the ratio of noisy instance $\alpha$ increases. These results confirm the effectiveness of iFlood in denoising.

### F.4 CONFIDENCE DISTRIBUTION

To further confirm that "iFlood encourages the model to better fit the under-fitted instances while suppressing the confidence of over-fitted ones" from the perspective of model confidence, we demonstrate the distribution of model confidence in Figure 7. We adopt the same experimental settings as those used in Figure 1: a ResNet-18 is trained on CIFAR-10, and the flooding level $b$ is set to $0.03$ for both Flooding and iFlood. From the figure we can observe that, compared with Flooding, iFlood encourages the model to continue to fit the under-fitted instances w.r.t. $b$, (i.e., the instance with confidence less than $e^{-b} \approx 0.97$), while suppressing the over-fitted ones (i.e., the instance with confidence large than $e^{-b} \approx 0.97$). These results are consistent with those in Figure 1.

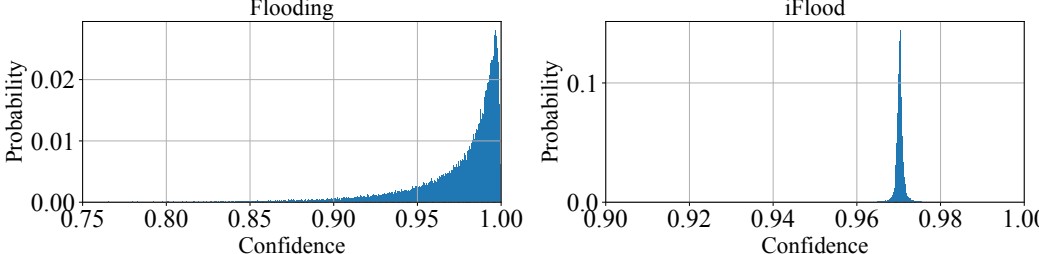

Figure 7: The distributions of model confidence.

