# OpenReview forum: "iFlood: A Stable and Effective Regularizer"
_ICLR.cc/2022/Conference — ICLR 2022 Poster_

### Official Review · Reviewer_bQVY · 2021-11-02

**Correctness:** 3
**Technical Novelty And Significance:** 3
**Empirical Novelty And Significance:** 2
**Recommendation:** 6
**Confidence:** 4

**Details Of Ethics Concerns:**

No ethics concerns

**Main Review:**

Strengths:
+ This paper is well written and it is enjoyable to read.
+ The idea of iFlood is neat.
+ It is theoretically showed that the design of iFlood can be intrinsically connected with removing the noise or bias in training data.

My main concerns are on the effectiveness of iFlood:
+ There are many regularizations which can prevent overfitting. I think more powerful baselines should be added (e.g. mix up).
+ The improvements are very marginal. This makes me very doubt whether iFlood is really effective.
+ The performance of baseline in GLUE is not convincing (SST-2 and MNLI). It is shown in https://github.com/huggingface/transformers/blob/master/examples/pytorch/text-classification/README.md that sst-2 can achieve 93.2 and mnli can achieve 83.9.


**Summary Of The Paper:**

This paper proposed a stable and effective regularizer to prevent overfitting. Specifically, this paper proposes individual Flood. Different from Flood which constrains training loss on mini-batch level, iFlood gives instance-level constraints on training loss. This paper also theoretically shows that the design of iFlood can be intrinsically connected with removing the noise or bias in training data.

**Summary Of The Review:**

I think the idea of iFlood is neat and simple. Theoretical analysis is good enough. However, the effectiveness of iFlood can not be guaranteed in experiments. So I think this paper is marginally below the acceptance threshold.

---

> ### Author Response · Authors · 2021-11-18
> **Thank you for the detailed comments! Please see our responses below.**
>
> Many thanks for your comments! We address your concerns accordingly:
>
> 1. For comments on **more baselines in performance comparison**:
> Thanks for your suggestions on experiments.
> (a) To confirm the effectiveness of iFlood, we compare iFlood with more regularizers, including Mixup[1] and DropBlock[2] (a modified version of Dropout for CNN). We adopt DropBlock rather than vanilla Dropout since DropBlock is empirically proven to be more suitable for ResNet, which provides a stronger baseline.
> &emsp;&emsp;&emsp;Table 1. Accuracy (%) comparison on benchmark datasets.
> | Datasets | Mixup | DropBlock | iFlood | iFlood+Mixup | iFlood+DropBlock|
> |  :----  | :----:  | :----: | :----: | :----: | :----: |
> | CIFAR-10 | 95.02 | 94.79 | 94.95 | 95.81 | 95.05 |
> | CIFAR-100 | 77.45 | 77.73 | 79.06 | 78.25 | 77.95 |
> | SVHN | 97.10 | 96.93 | 97.16 | 97.20 | 97.23 |
>
>     The experimental results are shown in the above table, from which we can observe that iFlood achieves comparable performance on CIFAR-10, and outperforms baseline models on CIFAR-100 and SVHN. The experimental results confirm the effectiveness of iFlood.
> (b) Further, although various regularizers have been proposed to prevent overfitting, such as Weight Decay, Batch Norm, Mixup, etc., iFlood provides another solution from a new perspective: directly controlling the individual training loss. And since iFlood is orthogonal to the existing regularizers, these regularizers can collaboratively work under suitable configurations. For example, we empirical show that iFlood works well with Mixup and DropBlock. We can observe that on CFIAR-10, iFlood+Mixup v.s. Mixup = 95.81 v.s. 95.02 (+0.79%), and iFlood+DropBlock v.s. DropBlock = 95.05 v.s. 94.79 (+0.26%).
>
>
> 2. For comments on **the effectiveness of iFlood**:
> (a) Compared to related works, models learned with iFlood have achieved a similar and undoubtedly noticeable outperformance. For example, according to the reported results in the original paper of Flooding [3], the largest improvements brought by Flooding on CIFAR-100 and SVHN are 0.32% and 0.43% respectively, and those of iFlood are 0.82% and 0.22%. And according to the reported results in the original paper of Confidence Penalty [4], Label Smoothing and Confidence Penalty improve 0.15% and 0.27% respectively on CIFAR-10 in terms of accuracy, while iFlood improves 0.36%. These improvements confirm the effectiveness iFlood.
> (b) We further theoretically show that the design of iFlood can be intrinsically connected with removing the noise or bias in training data, which makes it suitable for a variety of applications to improve the generalization performances of learned models. And we conduct a series of experiments to provide empirical evidence to confirm the effectiveness of iFlood.
>
>
> 3. For comments on **the performance of baselines in GLUE**:
> Thanks for your comments! The reasons for the differences between the reported results in Table 4 in Appendix D.2 and the board released by Huggingface includes:
> (a) The most important reason is that the results reported in the mentioned link are obtained by **a single training**, which contains randomness. And our experimental results are obtained by **running models 5 times** to reduce the randomness.
> (b) We conduct hyperparameter optimization to search for optimal configuration. Some hyperparameters might be different, such as the learning rate (search from {2e-5, 5e-5} v.s. 2e-5), number of epochs (search from {3,4,5} v.s. 5 for MRPC and WNLI, 3 for others). And the batch size is set to 16. More details of the implemental details for reproduction can be found in Sec.4.1 and Appendix B.
> What's more, it is worth pointing out that, except for SST-2 and MNLI, the baselines achieve better performance in our submission than those results released by Huggingface on other 6 datasets (and equivalent results on WNLI).
> In summary, we believe and argue that the GLUE results reported in this submission are convincing and reproducible. And similar results are also achieved by previous studies that implement vanilla BERT based on Huggingface. For example, the reported result of STS-2 in [5] is 91.50 (91.88 in our submission), and the reported result of MNLI in [6] is 83.16 (83.22 in our submission).
>
>
> Thanks again for your detailed comments! Besides, we respectfully ask to consider increasing the rating score if our clarifications have addressed the concerns you raised.
>
>
> Ref:
> [1] Mixup: Beyond Empirical Risk Minimization. *ICLR, 2018*.
> [2] DropBlock: A Regularization Method for Convolutional Networks. *NeurIPS, 2018*.
> [3] Do We Need Zero Training Loss After Achieving Zero Training Error? *ICML, 2020*.
> [4] Regularizing Neural Networks by Penalizing Confident Output Distributions. *ICLR, 2017*.
> [5] BERT-of-Theseus: Compressing BERT by Progressive Module Replacing. *EMNLP, 2020*.
> [6] EarlyBERT: Efficient BERT Training via Early-bird Lottery Tickets. *ACL, 2021*.

---

> > ### Comment · Reviewer_bQVY · 2021-11-19
> > **Your answers are convincing.**
> >
> > The authors' rebuttal addresses a number of my comments. The new results are great and show the effectiveness of iFlood. The reasons for the differences in GLUE are also convincing for me. These are good additions for a future version. Thus I would recommend the authors incorporate the rebuttal content into the next version. I have increased my ratings.

---

### Official Review · Reviewer_asHY · 2021-11-02

**Correctness:** 4
**Technical Novelty And Significance:** 4
**Empirical Novelty And Significance:** 3
**Recommendation:** 6
**Confidence:** 4

**Details Of Ethics Concerns:**

I think there are no ethics concerns.



**Main Review:**

This paper is well organized and easy to read. Although the method is simple, it has a specific theoretical basis.
My primary concerns are listed as follows:

1. In Section 3.1, the authors gave some analyses and concluded that ''iFlood encourages the model to better ﬁt the under-ﬁtted instances while suppressing the conﬁdence of over-ﬁtted ones.'' However, according to the analysis, it is difficult to determine how this conclusion was reached. Since, sometimes, the individual loss cannot reflect the model confidence. I think more experiments about model calibration are needed to support this claim.

2.  In Eq.7, why do the authors use the Laplace distribution to simulate the loss function?

3. For Table 2, the authors provided the results of variances. I understand that the authors evaluate ten models trained with different methods. I would like to know whether these results were obtained under fixed random seeds? Or is the random seed selected randomly for each experiment evaluation?

4. In Figures 2 (a) and (b), I noticed some abrupt changes in the learning curve, especially near the second drop of the learning rate. What is the reason for this phenomenon?

5. I think the experiments of label noise are not very good compared to current methods (like self-adaptive training [1]).

[1] Self-Adaptive Training: beyond Empirical Risk Minimization, NeurIPS 2020.





**Summary Of The Paper:**

This paper introduced a simple but effective method to mitigate overfitting, which modifies the existing Flooding scheme. The proposed iFlood can solve the potential problems of the Flooding algorithm and improve the stability. Furthermore, the authors gave some theoretical analyses of the proposed method. The experiments indicated the proposed method is better than the baselines.


**Summary Of The Review:**

I think the proposed iFlood is simple but effective. The theoretical analysis is straightforward. However, there are still some problems, and the experimental results are not good. So, I recommend to ''marginally below the acceptance threshold''.

---

> ### Author Response · Authors · 2021-11-18
> **Thank you for the detailed comments! Please see our responses below.**
>
> Many thanks for your thoughtful comments! We have revised our paper following the suggestions and responded to your questions accordingly:
>
> 1. For comments on **model calibration**:
> Many thanks for your comments!
> (a) The loss function of iFlood encourages the individual losses to be approached to $b$, i.e., $|L_i-b|\approx 0, \forall i \in \[N\]$. For $L_i > b$, i.e., the under-fitted instances, models are encouraged to continue to fit them via contributing $\nabla_{\theta} L_i$; while for $L_i < b$, i.e., the over-fitted instances w.r.t. $b$, models are encouraged to contribute the negation of gradient $- \nabla_{\theta} L_i$. Therefore we claim that "iFlood encourages the model to better fit the under-fitted instances while suppressing the confidence of over-fitted ones".
> (b) To further confirm it from the perspective of model confidence, we plot the distribution of model confidence in Figure 7 (please refer to Appendix F.4). From the figure we can observe that, compared to Flooding, iFlood encourages the model to continue to fit the under-fitted instances w.r.t. $b$, (i.e., the instance with confidence less than $e^{-b}\approx 0.97$), while suppressing the over-fitted ones (i.e., the instance with confidence large than $e^{-b}\approx 0.97$). The results are consistent with those in Figure 1 in the submission, which supports our conclusion.
> (c) Moreover, we conduct the suggested experiments to show the effect of iFlood in calibration. Following [1], we adopt the Expected Calibration Error (ECE) as the empirical metric to evaluate the model calibration. The bins number is set to 10, and smaller ECE value is better.
> &emsp;Table 1. Expected Calibration Error (%) on CIFAR-10 and CIFAR-100.
> | | Unreg. | Flooding | iFlood |
> |:----| :----: | :----: | :----: |
> | CIFAR-10 | 3.59 | 2.17 | 1.75 |
> | CIFAR-100 | 9.76 | 7.09 | 3.21 |
>
>     The experimental results are shown in the above table, from which we can observe that model learned with iFlood can achieve a better result on model calibration. Inspired by [2] and the relationship between iFlood and Label Smoothing discussed in Sec.3.4, we conjecture that iFlood can implicitly calibrate models’ predictions as what label smoothing does.
> Thanks again for your valuable comments!
>
>
>
> 2. For comments on **Laplace distribution**:
> Thanks a lot for your comment! In some real-world scenarios, the distance (e.g., Cross-Entropy) between ground-truth label and biased label obeys a single-sided long-tailed distribution deviating from a small positive value (serving as our $b$), which motivates our assumption (i.e., Eq.(7)). Although might not be perfect, it helps to understand when and why iFlood works for debiasing.
>
>
> 3. For comments on **random seeds**:
>  The reported results are obtained under fixed random seeds for a fair comparison.
>
>
> 4. For comments on **the changes in learning curve**:
>  We agree that the changes are caused by the decay of learning rate at this epoch. As shown in Figure 6 in Appendix F.2, we observe that the changes in learning curve happen when models are learned with different regularizers on different datasets. Based on these experimental results, we believe that such phenomena are not related to the adopted regularization methods. The deeper reasons might be related to the optimization landscape and beyond the scope of this study.
>
>
> 5. For comments on **the experiments of label noise**:
>  Thank you for your comment!
> The goal of providing the analysis from the perspective of denoising is to better understand how iFlood works, while improving the performance for denoising is not the primal goal of this study.
>
>
> Thanks again for your detailed comments! Besides, we respectfully ask to consider increasing the rating score if our clarifications have addressed the concerns you raised.
>
>
> Ref:
> [1] On Calibration of Modern Neural Networks. *ICML, 2017*.
> [2] When Does Label Smoothing Help? *NeurIPS, 2019*.

---

> > ### Comment · Reviewer_asHY · 2021-11-24
> > **Thanks for your response.**
> >
> > The new results of model calibration is convince. The response has addressed most of my comments, I'd like to increase my score.

---

### Official Review · Reviewer_WRr1 · 2021-11-02

**Correctness:** 3
**Technical Novelty And Significance:** 4
**Empirical Novelty And Significance:** 3
**Recommendation:** 6
**Confidence:** 4

**Main Review:**

Strengths:
- Idea is simple and novel.
- The empirical improvement compared to standard flooding is encouraging.
- Experiments on noisy labels show significant improvement.

Weaknesses:
- The formulation of noisy labels and biased sample distribution is unclear. The current formulation of biased sample seems like another form of incorrect labels, where P(z|x) and P(y|x) are different. These two forms of noisy labels could potentially be merged in a unified setting. On the other hand, I am wondering whether distribution shift in the input distribution P(x) could be covered as well?
- I believe the benefit of noisy labels and biased samples should be further investigated. Currently only 2 out of the 7 datasets ran a simple noisy label experiment. There is no empirical evaluation of biased samples since the noisy labels are just uniform (based on the description in Appendix E).
- It’s unclear whether the stated theoretical properties applies only to iFlood or both Flood and iFlood and I would appreciate it if the authors can clarify the relations here.
- On standard benchmarks the improvement seems marginal. This wouldn’t be a major problem if the other properties such as stability and noisy labels can be highlighted more in the experiments.
- Regarding to the claims around stability, the paper uses several measures to look at the loss distribution and the gradient norm. However, there isn't strong and direct evidence that Flooding leads to "unstable" training, i.e. the training loss diverges. This argument could be strengthened more.

**Summary Of The Paper:**

The paper proposes a flooding loss function that encourages the training loss for each example to be a positive bias instead of zero. The paper has found that such training objective stabilizes the training compared to the regular Flooding, which regularizes the average loss instead. The paper also provides reasoning why such a loss function provides more robustness for training with noisy labels and biased label distributions. The empirical benefit on standard datasets seems marginal, but the benefit on noisy labels is significant.

**Summary Of The Review:**

This paper presents an interesting loss function that provides more robustness and also performs slightly better in regular settings. I found the paper interesting and potentially have a strong impact. I think the experimental design and theoretical sections could be strengthened more and highlight more benefits dealing with noisy labels and dataset biases. My overall rating is “weak accept”.

---

> ### Author Response · Authors · 2021-11-18
> **Thank you for the detailed comments! Please see our responses below.**
>
> Many thanks for your thoughtful comments! We have revised our paper following the suggestions and addressed all of your comments in the following response:
>
> 1. For comments on **the formulation of noisy label and biased sample**:
> Many thanks for your thoughtful comments.
> (a) The most important difference between noisy label and biased sample is that, in noisy label setting, we assume that the observed noisy instance {$(x_i, z_i)$} is drawn from $\Pr(X, Z)$, which differs from the ground-truth joint distribution $\Pr(X, Y)$ as $\Pr(Z|X)$ differs from $\Pr(Y|X)$. However, in biased sample setting, although the joint distribution $\Pr(X, Z)$ is also different from $\Pr(X, Y)$, the bias lies in the distribution of input. That is, even $\Pr(Z|X) = \Pr(Y|X)$, $\Pr(X, Z)$ can still differ from $\Pr(X, Y) $ as $\Pr(Z, X) = \Pr(Z|X)\Pr'(X) \neq \Pr(Y|X)\Pr(X)=\Pr(X, Y)$.  The observed distribution of input, denoted as $\Pr'(X)$, is different from the ground-truth one $\Pr(X)$.
> (b) Therefore, the biased sample can cover the distribution shift in the input distribution as shown above.
> We have added the analysis above in the revised version.
> Thanks again for your comments!
>
>
> 2. For comments on **noisy label experiments**:
> Many thanks for your comments. The goal of the experiments on noisy label is to further confirm the theoretical analysis in Sec 3.3.1, therefore the noisy labels follow the uniform distribution. Although such setting is not perfect and cannot cover lots of different scenarios, it helps to understand why iFlood works for denoising.
>
>
> 3. For comments on **the theoretical properties**:
> The theoretical analysis in the submission is only applied to iFlood, with the aim to analyze when and why do the models learned with iFlood benefit from the regularization.
> And the **relations** between iFlood and Flooding can be summarized as: The optimal solution to minimize the loss function of iFlood $\mathcal{L}=\frac{1}{N} \sum_{i=1}^{N} \big(|L_i-b|+b\big)$ is $L_i=b, \forall i \in [N] $, which is also one of the feasible solutions to minimize the loss function of Flooding $|\big(\frac{1}{N} \sum_{i=1}^{N} L_i\big)-b|+b$.
>
>
> 4. For comments on **stability**:
> Thanks for your valuable comments.
> As pointed out on Page 2 of the submission, the meaning of "instability" (or "unstable") is different from that used in learning theory [1], and does not relate to the divergent training loss. This is because we are studying the topic of regularization and generalization, and thus it is more important to ensure the stability in terms of solution quality. To this end, we investigate how models learned with Flooding behave and uncover the instability issue of it, which means that it can lead to different solutions, and the solutions are inconsistent in their generalization abilities and their behaviors over individuals.
>
>
> Thanks again for your detailed comments! Besides, we respectfully ask to consider increasing the rating score if our clarifications have addressed the concerns you raised.
>
>
> Ref:
> [1] Foundations of machine learning. *MIT press, 2018*.

---

> ### Author Response · Authors · 2021-12-03
> **Thanks again for the detailed comments! Looking forward to your feedback.**
>
> Dear Reviewer WRr1,
>
> Thanks again for your thoughtful comments!
> As the discussion period is close to the end, we would appreciate it if you could kindly let us know the response has addressed your concerns. And we are happy to discuss further if you have any questions.
>
> Authors

---

### Official Review · Reviewer_fRos · 2021-11-02

**Correctness:** 3
**Technical Novelty And Significance:** 3
**Empirical Novelty And Significance:** 2
**Recommendation:** 6
**Confidence:** 4

**Main Review:**

**Strength**

The paper poses several interesting contributions in terms of training of models while limiting the models' overly attention to a subset of samples.
 - The idea of extending Flooding to iFlooding is very intuitive, thus it's fairly clear that such extension is needed.
 - Some additional discussions on the properties iFlooding in terms of robustness to noise labels and biased samples are offered.

**Weakness**

On the other hand, I also have several concerns about the presentation of the idea, especially on the empirical end
 - It's not clear that in the experiments, whether each data are shuffled when every iteration ends. Intuitively speaking, SGD (and its variants) has a natural counter to overly attending specific samples as the gradient are calculated every batch and shuffling samples at every iteration will avoid the batch to stay the same, thus the gradient will always be different (even if the weights are the same), avoiding some overly attending to specific samples. Therefore, the lack specification of this setup posts a challenge to appreciate the effects of the empirical results.

 - Related, it has been shown that when the model is being trained, the model pays attention to different levels of details of the data as the iteration goes (Figures 4, 5 in [1]), therefore, instead of presenting a fixed table at the end of all the iterations, showing the curves of accuracies along the training process will help better appreciate the idea of iFlooding. It might also be interesting to present accuries on different test copies as in (Figures 4, 5 in [1]).

 - If I understand correctly, Table 1 is a comparison of generalization (accuracy) improvement ability of methods, instead of a comparison of the methods devoted to calibrating the overly attention, thus conventional generalization boosting methods such as dropout, batchnorm, or even mixup should probably also been compared. I don't think the authors need to worry whether BatchNorm will deliver a higher accuracy than what their method gets, since these methods are motivated by different problems. The authors could always discuss these, or present the numbers in different cells. However, my point is that for a table that serves as a comparison of accuracy-boosting techniques, it's probably not a great idea that these conventional methods are not compared.

 - I do not fully follow why Sections 4.3 and 4.4 are devoted only to iFlooding and Flooding, why it is not interesting to compare other methods? Even if these methods do not explicitly account for stability, it's still interesting to see the curves, especially since it seems Figure 2(a)(b) can be computed simultaneously for the experiments used to generate Table 1 with no additional computing resources needed.

I might have asked for too many empirical results, which may not be doable in the rebuttal phase, but I will suggest the authors at least touch each of these with the CIFAR10 experiment.

[1] High Frequency Component Helps Explain the Generalization of Convolutional Neural Networks

**Summary Of The Paper:**

This paper extends the idea of Flooding to a sample-specific level and call it iFlooding. The extension is intuitively important and the authors also offered several analytical discussions to show its importance beyond the intuition. The empirical results are fairly relevant and strong.

**Summary Of The Review:**

I think the paper is very interesting, intuitive, and a natural extension of the Flooding idea. The empirical section can be improved though.

---

> ### Author Response · Authors · 2021-11-18
> **Thank you for the detailed comments! Please see our responses below.**
>
> Many thanks for your thoughtful comments and valuable suggestions! We have revised our paper following the suggestions and addressed your concerns in the following:
>
> 1. For comments on **data shuffled**:
> For all the experiments conducted in our study, the data are shuffled when every iteration ends. And we absolutely agree that such a shuffling mechanism can be regarded as an implicit regularization to prevent overfitting, therefore it has been applied to all the baseline methods we adopted for a fair comparison.
>
>
> 2. For comments on **showing the curves of accuracy**:
> Many thanks for providing a new perspective to analyze iFlood.
> Following your suggested reference [1], we conduct additional experiments to show how model learned with different regularizers reacts to different levels of details of the data (e.g., the low-frequency component and high-frequency component of images). These experiments are illustrated in Figure 5 (Please refer to Appendix F.1 in the revised version). From these results we can observe that model learned with iFlood catches more low-frequency component and less high-frequency component than Vanilla (i.e., the models learned without Flooding and iFlood) and Flooding, which confirms that iFlood can help to capture generalizable patterns.
>
>
> 3. For comments on **the comparison of generalization**:
> Thanks a lot for your valuable comments on the comparison of generalization!
> To further confirm the effectiveness of iFlood, we compare iFlood with more regularizers, including Mixup[2] and DropBlock[3] (a modified version of Dropout for CNN). We adopt DropBlock rather than vanilla Dropout since DropBlock is empirically proven to be more suitable for ResNet. As for BatchNorm, we have adopted it in ResNet for both iFlood and baseline methods, since BatchNorm can be regarded as the necessary plugin for achieving competitive performance. And the experiments in the submission demonstrate that iFlood works well together with BatchNorm.
> &emsp;&emsp;&emsp;Table 1. Accuracy (%) comparison on benchmark datasets.
> | Datasets | Mixup | DropBlock | iFlood | iFlood+Mixup | iFlood+DropBlock|
> |  :----  | :----:  | :----: | :----: | :----: | :----: |
> | CIFAR-10 | 95.02 | 94.79 | 94.95 | 95.81 | 95.05 |
> | CIFAR-100 | 77.45 | 77.73 | 79.06 | 78.25 | 77.95 |
> | SVHN | 97.10 | 96.93 | 97.16 | 97.20 | 97.23 |
>
>     The experimental results are shown in the above table, which confirm the effectiveness of iFlood.
> Note that different regularizers are motivated from different perspectives to prevent overfitting, and they can collaboratively work under suitable configurations.
> Therefore, we empirical show that iFlood works well with Mixup and DropBlock. For example, on CFIAR-10, iFlood+Mixup v.s. Mixup = 95.81 v.s. 95.02 (+0.79%), and iFlood+DropBlock v.s. DropBlock = 95.05 v.s. 94.79 (+0.26%).
>
>
> 4. For comments on **comparing to more baselines in Section 4.3 and 4.4**:
> Many thanks for your suggestions!
> (a) The goal of the comparisons in Section 4.3 and 4.4 is to better understand how the design of loss function of iFlood can work well as we expected.
> (b) For further analysis, we conduct the suggested experiments and show the results in Figure 6 (Please refer to Appendix F.2 and F.3). These results confirm that (i) iFlood significantly reduces the norm of gradients during the training course and leads to a stable generalization boost. (ii) Models learned with iFlood outperform those with other regularizers by a noticeable margin on polluted dataset, which confirms the effectiveness of iFlood in denoising.
>
>
> Thanks again for your detailed comments! Besides, we respectfully ask to consider increasing the rating score if our clarifications have addressed the concerns you raised.
>
>
> Ref:
> [1] High-frequency Component Helps Explain the Generalization of Convolutional Neural Networks. *CVPR, 2020*.
> [2] Mixup: Beyond Empirical Risk Minimization. *ICLR, 2018*.
> [3] DropBlock: A Regularization Method for Convolutional Networks. *NeurIPS, 2018*.

---

> > ### Comment · Reviewer_fRos · 2021-11-18
> > **Thanks for the clarification**
> >
> > Thank you for the clarification. The response has addressed most of my concerns, and I'm happy to update my ratings.

---

### Decision · Program_Chairs · 2022-01-20

**Decision:**

Accept (Poster)

**Comment:**

The paper extends the original work on flooding to individual instance level to prevent overfitting. Even though the technique is a intuitive extension, the reviewers appreciate its simplicity and effectiveness, and consider the extension necessary. Most reviewers' concerns were addressed through rebuttal.